# DIFFERENTIABLE, STABLE AND EFFICIENT FLOATING-POINT QUANTIZATION

## ABSTRACT

Finding optimal datatype for neural networks is a non-trivial problem with exponential search space. To solve the problem of quantization effectively, we consider pseudo-quantization training (PQT) on microscaling (MX) datatypes. Specifically, we propose pseudo-quantization noise (PQN) based on $R \approx \lfloor \mathcal{N}(0,1)/2 \rceil$. It allows PQT to (1) optimize on the floating-point (FP) bit configuration, (2) help preserve dynamic range of original data, and (3) generate noise $R$ efficiently. We demonstrate that the proposed method allows for stable and efficient pre-training of the GPT2 and Llama2 language models up to 1 billion (B) parameters for up to 295B tokens, with insights on optimal FP datatypes for model parameters.

## 1 INTRODUCTION

Quantization has been a promising solution for efficiency. However, it is a non-trivial problem with at least 2 degrees-of-freedom where we need to decide the datatype and the range of values to be represented with the datatype. While microscaling (MX) datatypes (Project, 2023) fix the quantization range as adaptive power-of-two and reduce the degree-of-freedom down to 1, the problem search space easily becomes exponential when we want to find optimal mixed-precision datatype. For example, deciding layerwise datatypes from {MXFP8_e4m3, MXFP4_e2m1}[1] for a $n$-layer model yields $O(2^n)$ cases. Besides, large language models (LLMs) require >200B tokens of training for each case to ensure training stability (Fishman et al., 2025).

Pseudo-quantization training (PQT) effectively reduces the search space down to $O(1)$ by employing differentiable pseudo-quantization noise (PQN) as a generalization over actual quantization noise. For example, the formulation of DiffQ is $\widehat{W} = W + R \cdot \Delta$ where $R \sim U(-0.5, 0.5)$ and $\Delta$ is the stepsize for $B$-bit integer, defined as $\frac{\max(W)-\min(W)}{2^B-1}$. Note that the formula is fully differentiable to allow optimization on the bitwidth parameter $B$. It also promotes stable training by regularizing Hessian diagonal of loss (Shin et al., 2023). However, existing PQT methods focus exclusively on integer datatypes for inference and overlook numerical behavior of noise addition during training.

We address the problem of determining MXFP datatypes of model parameters, both for inference and training, via PQT. Our contributions with the proposed method, *i.e., DiffFPQ*, are as follows:

- We employ rounded normal $R \sim \lfloor \mathcal{N}(0,1)/2 \rceil$ as a basis of PQN. It extends PQT to target MXFP while preserving the dynamic range of the original model parameter $W$.

- We demonstrate stable PQT that closely follows, or even outperforms, the baseline BF16 on pre-training GPT2-124M and Llama2-{134M, 1B} language models up to 295B tokens.

- We demonstrate efficient PQT with a 3.14% overhead in training throughput on the A100 GPU by leveraging bitwise operations (bitops) to generate the approximated distribution $R \approx \lfloor \mathcal{N}(0,1)/2 \rceil$.

- We demonstrate that *DiffFPQ*-trained models achieve Pareto-optimal benchmark results, especially with mixed-precision parameters $W$ in {FP12_e4m7, FP8_e3m4, FP4_e2m1} following the resulting bitwidth parameter $B_t$.

- We further demonstrate that *DiffFPQ* can suggest a baseline datatype, *i.e.,* the datatype for stable training, of the model parameter.

---

[1] (MX)FPn_eEmM represents n-bit (MX) floating-point with E-bit exponent and M-bit mantissa in this paper.

## 2 RELATED WORK

Microscaling (MX) has been proposed as an open compute project (OCP) standard (Project, 2023). Quantization with MX separates given data into multiple blocks where low-precision elements within a single block share a single power-of-two scale. It effectively separates dynamic range of input data from tensor-wise down to inter- and intra-block where intra-block quantization range is adaptive power-of-two. That is, the problem of quantization becomes disaggregated: all we need to consider is intra-block representation ability where the quantization range is given as power-of-two.

DiffQ (Défossez et al., 2022) proposed pseudo-quantization noise (PQN) to train bitwidth parameters with fixed quantization range. NIPQ (Shin et al., 2023) proposed pseudo-quantization training (PQT) that trains both bitwidth parameter and quantization range. It also proved that such training converges to minima which is flatter by implicitly regularizing the Hessian diagonal term of the loss. Note that existing PQT works consider integer datatypes while MX datatypes are mostly FP.

Fully quantized training (FQT), quantization-aware training (QAT) and post-training quantization (PTQ) are well-known methodologies when it comes to quantization. However, existing quantization works except for PQT are inefficient when it comes to mixed-precision quantization. They either mandate exponential search as discussed in Section 1, or require $O(n)$ time-resource as in neural architecture search where $n$ is the number of options for each layer (Nair et al., 2025). In contrast, PQT directly optimizes the bitwidth parameter through gradient descent with constant overhead.

## 3 METHOD

We consider stable and efficient PQT that targets MXFP parameters. Section 3.1 introduces the PQT formulation that targets MX datatypes. In Section 3.2, we analyze the implications of PQN addition and propose the rounded Gaussian $R \sim \lfloor \mathcal{N}(0,1)/2 \rceil$ to allow stable PQT that targets MXFP parameters. Section 3.3 introduces efficient generation of the approximated rounded normal distribution $R \approx \lfloor \mathcal{N}(0,1)/2 \rceil$ by employing bitwise operations (bitops) in place of FP operations. Section 3.4 explains design choices that favor predictably optimal throughput and modular implementation for the Triton (Mattson et al., 2019)-based GPU kernels. Section 3.5 describes implementation details, including the method to ensure forward-backward consistency with unbiased PQN, and the way to implement bitwidth parameter.

### 3.1 FORMULATION

We assume a square-blockwise quantization to target MX datatypes. Unlike vector-wise quantization, square-blockwise quantization guarantees transpose-commutativity which is essential for stable training. Refer to Appendix C and Chen et al. (2025) for detail. Furthermore, it remains MX-compliant, as square-blockwise quantization can be viewed as a special case of vector-wise quantization where adjacent vectors share a common scale.

The formula that we consider is:

$$\widehat{W} = W + R \odot \text{broadcast}_{b_l} \left( \max_{b_l}(|W|) \cdot 2^{1-B_t} \right) \tag{1}$$

where $\{W, \widehat{W}, R\} \in \mathbb{R}^{m \times n}$, $B_t \in \mathbb{R}^{\lceil m/b_l \rceil \times \lceil n/b_l \rceil}$, and $b_l = 32$ is the square block size following MX. $W$ and $\widehat{W}$ denote the original and sampled parameters, respectively. $R$ represents random and $B_t$ is blockwise bitwidth where $b_t$ represents an element of it. $\max_{b_l}$ denotes square-blockwise maximum while $\text{broadcast}_{b_l}$ is a function $f : \mathbb{R}^{(m/b_l) \times (n/b_l)} \to \mathbb{R}^{m \times n}$ that replicates the same value square-blockwise. $\odot$ and $\cdot$ denote the Hadamard product, $|\cdot|$ denotes the elementwise absolute. We refer to the right-hand side of the addition as PQN.

Note that Equation 1 is fully differentiable. With an approximation of $\frac{\partial \max_{b_l}(|W|)}{\partial W} \approx 0$ assuming gradient to single element out of 32 by 32 block is negligible, we can calculate the gradient with respect to the target loss $\mathcal{L}$ as follows:

$$\frac{\partial \mathcal{L}}{\partial W} = \frac{\partial \mathcal{L}}{\partial \widehat{W}} \quad \text{and} \quad \frac{\partial \mathcal{L}}{\partial B_t} = -\ln 2 \cdot \max_{b_l}(|W|) \cdot 2^{1-B_t} \cdot \sum_{b_l} \left( \frac{\partial \mathcal{L}}{\partial \widehat{W}} \odot R \right) \tag{2}$$

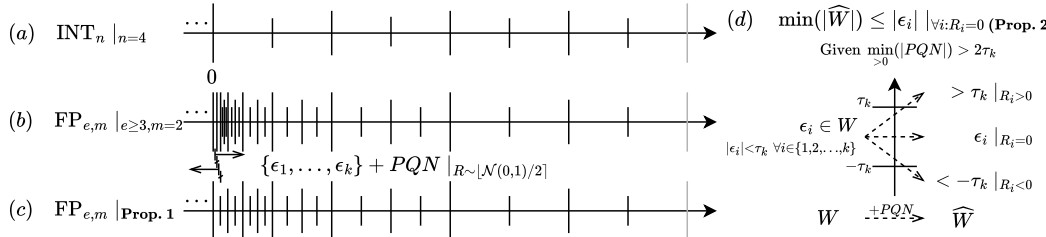

Figure 1: Representable values of power-of-two-scaled integer and floating-point datatypes, with respect to adding PQN to near-zero elements $\epsilon_i$. Comparing $(a)$ to $(b)$ shows that FP with $(n-2)$-bit mantissa includes $n$-bit integers. $(c)$ demonstrates reduced effective resolution of near-zero elements when adding non-zero elements of PQN. Non-trivial $P(R = 0)$ allows stochastic precision annealing of near-zero elements as a mixup of $(b)$ and $(c)$. It also encourages preserving the minimum magnitude of original data as $(d)$ demonstrates.

### 3.2 Implication of adding pseudo-quantization noise (PQN)

While PQN addition in Equation 1 seems lossless, it implies FP casting and loses information of the smaller-magnitude operand. In this section, we discuss the implication of PQN addition and FP casting to propose $R \sim \lfloor \mathcal{N}(0,1)/2 \rfloor$ which allows stable PQT that targets FP datatypes.

**Notation.** We consider $fp_{e,m}(\widehat{W})$ where $fp_{e,m}(\cdot)$ is casting to FP with $e$-bit exponent and $m$-bit mantissa. For MX extension of it, for simplicity without loss of generality, we consider a single MX block with a scalar $b_t \in [3, 12]$ since PQN and MX version of $fp_{e,m}(\cdot)$ share granularity. For arbitrary block, consider scalar elements $R_i$, $PQN_i$ and $\widehat{W}_i$ that corresponds to $\epsilon_i = W_i \in W$, and minimum positive PQN $\epsilon_{PQN} = \max(|W|) \cdot 2^{\rho+1-b_t}$ where $2^\rho \triangleq \min_{>0}(|R|)$ with $\min_{>0}(X) \triangleq \min(X : X_i > 0, \forall i)$. We define the resolution of scalar FP $x$, denoted as $\Delta_{\min}(x)$, as the minimum non-zero delta of $x$. Specifically, $\Delta_{\min}(x) = 2^{\lfloor \log_2(|x|) \rfloor - m}$ given that $|x|$ is in normal range $[2^{\lfloor \log_2(|x|) \rfloor}, 2^{\lfloor \log_2(|x|) \rfloor + 1})$ and $m$ is the number of FP mantissa bits.

**Stochastic precision annealing.** As resolution of FP depends on magnitude, addition potentially loses information of the smaller-magnitude operand. For example, resolution of $\epsilon_i$ is lost during $\epsilon_i \pm \epsilon_{PQN}$ as shown in Figure 1. It suggests that adding PQN limits the resolution of near-zero elements, to limit the effective number of FP exponent bits.

**Proposition 1.** *Assume PQN that corresponds to $b_t$-bit with $2^\rho \triangleq \min_{>0}(|R|)$ and $P(R = 0) \approx 0$. FP addition $\widehat{W} = W + PQN$ leads to limited representation of $W$ so that floating-point with a $\lceil \log_2(-\rho + b_t + 1) \rceil$-bit exponent suffices to represent $W$.*[2]

Figure 1(c) demonstrates limited representation of $W$. Now consider $P(R = 0)$. $R_i = 0$ leads to $PQN_i = 0$ and $\widehat{W}_i = W_i$. It effectively enables stochastic pass-through of high precision $W_i$ into PQT, only to be limited by $fp_{e,m}(\cdot)$. That is, non-trivial $P(R = 0)$ yields stochastic mixup of high precision $fp_{e,m}$ with $P(R = 0)$ and low precision $fp_{e,m}|_{e=\lceil \log_2(-\rho+b_t+1) \rceil}$ with $P(R \neq 0)$. Therefore, PQT with non-trivial $P(R = 0)$ trains the model to be robust to low-precision FP while preserving precision of $W$ at the same time. We name this property *stochastic precision annealing*.

**Impact on dynamic range.** As an example of the impact of PQN addition on representation ability, consider the dynamic range of $\widehat{W}$ compared to that of $W$. Note that dynamic range of $\widehat{W}$ is defined as $\max(|\widehat{W}|)/\min_{>0}(|\widehat{W}|)$ and the impact on maximum magnitude is relatively trivial with $b_t \geq 3$. Therefore, we focus on the minimum magnitude of $\widehat{W}$—specifically $\min_{>0}(|\widehat{W}|) = \widehat{W}_i|_{PQN_i=0}$.[3]

**Proposition 2.** *Consider $k$ elements $\{\epsilon_1, \epsilon_2, \ldots, \epsilon_k\} \in W$, and $\tau_k$ such that $0 < |\epsilon_i| < \tau_k \ \forall i \in \{1, 2, \ldots, k\}$. Assuming $\epsilon_{PQN} > 2\tau_k$ and $P(R = 0) = p$,*

$$P\left(\min_{>0}(|\widehat{W}|) < \tau_k\right) \geq 1 - (1-p)^k \tag{3}$$

---

[2]The proofs of Propositions 1 and 2 are provided in Appendix A.

[3]$\min_{>0}(|\widehat{W}|) = \widehat{W}_i|_{PQN_i \neq 0}$ with $R \sim U(-0.5, 0.5)$ is less likely. Refer to Appendix B for detail.

Table 1: Number of exponent and mantissa bits for floating-point datatypes with respect to $b_t$ if the basis of PQN employs non-trivial $P(R = 0)$ with $\min_{>0}(|R|) = 1$, *i.e.,* $\rho = 0$. 'Datatype' refers to possible FP datatypes that support the given exponent and mantissa.

| $b_t$ | Exponent | Mantissa | Datatype |
|---|---|---|---|
| 3 | 2 | 1 | FP4_e2m1 |
| 4 | 3 | 2 | FP6_e3m2 |
| 5 | 3 | 3 | FP8_e4m3, FP8_e3m4 |
| 6 | 3 | 4 | FP8_e3m4 |
| 7 | 3 | 5 | BF16, FP16, FP12_e4m7 |
| 8 | 4 | 6 | BF16, FP16, FP12_e4m7 |
| 9 | 4 | 7 | BF16, FP16, FP12_e4m7 |
| 10 | 4 | 8 | FP16 |
| 11 | 4 | 9 | FP16 |
| 12 | 4 | 10 | FP16 |

As per Proposition 2, the minimum magnitude of $\widehat{W}$ is exponentially likely to be upper bounded by $\tau_k$ independent of $b_t$. Given fixed $k$, the condition favors preserving dynamic range of $W$ with higher $P(R = 0)$. That is, PQT with non-trivial $P(R = 0)$ favors preserving dynamic range of $W$, up to that of $fp_{e,m}(\cdot)$.

**Mantissa and choice of $R$.** With respect to the number of mantissa bits, we make the largest stepsize of the FP datatype equal to that of $b_t$-bit integer counterparts. Note that, as shown in Figure 1, FP with $(b_t-2)$-bit mantissa includes $b_t$-bit integer where 2 bits are compensated from FP standard: one for sign bit and the other for implicit leading 1 for values in normal range. Combined with the number of exponent bits from Proposition 1, PQT effectively targets $fp_{e,m}\mid_{e=\lceil \log_2(-\rho+b_t+1)\rceil, m=b_t-2}$.

We propose $R \sim \lfloor \mathcal{N}(0,1)/2 \rceil$ as a basis of PQN. First, it has a high probability of zero with $P(R = 0) \approx 0.68$. This allows PQT to be stable by preserving dynamic range of $W$, and to target FP datatypes of Table 1 through stochastic precision annealing. Second, it does not deviate largely from the previous works that proposed $U(-0.5, 0.5)$ and $\mathcal{N}(0,1)/2$ (Défossez et al., 2022; Shin et al., 2023). Lastly, approximated distribution $R \approx \lfloor \mathcal{N}(0,1)/2 \rceil$ can be generated efficiently.

## 3.3 EFFICIENT GENERATION OF $R$

Note that efficiency is critical for the proposed method, as LLMs frequently face throughput bottlenecks on CUDA cores—especially on datacenter GPUs like the A100. However, generating random numbers in the real number domain puts burden on CUDA cores by invoking FP operations on random bit streams produced by pseudo-random number generators (PRNGs) (Lathrop et al., 2011; Overton, 2020). For example, $U(0, 1)$ is derived by dividing the random integers by their maximum possible value. Two samples of $\mathcal{N}(0, 1)$ are derived from two samples of $U(0, 1)$ using the Box-Muller transform (Box & Muller, 1958).

Given discrete $R \approx \lfloor \mathcal{N}(0,1)/2 \rceil$, we can replace the aforementioned FP operations with bitwise operations (bitops) to achieve maximum efficiency during generation of $R$. Assuming each bit of the random integers generated by the PRNG is independently random, we can construct arbitrary discrete random distributions using two base cases:

$$\begin{cases} Pr(a \wedge b) = Pr(a) \cdot Pr(b) \\ Pr(a \vee b) = Pr(a) + Pr(b) - Pr(a \wedge b) \end{cases} \quad (4)$$

where $a$ and $b$ represent bitwise random variables, $\wedge$ and $\vee$ denote the logical-and and logical-or, respectively, and $Pr(x)$ is shorthand for $P(x = 1)$. Specifically, the distribution we generate is:

$$P(R_i = n) = \begin{cases} 3/4 \cdot 2^{-9} \approx 1/682.7 & \text{if } n \in \{-2, 2\} \\ (3/4)^2 \cdot 2^{-2} \cdot (1 - P(R_i = \pm 2)) \approx 1/7.1 & \text{if } n \in \{-1, 1\} \\ 1 - P(R_i = \pm 1) - P(R_i = \pm 2) \approx 0.717 & \text{if } n = 0 \end{cases} \quad (5)$$

In our implementation, the generated $R$ values are represented in a sign-mantissa format with 4 bits per element, and 8 elements are packed into a 32-bit register. Compared to 2's complement, the sign-mantissa format is simpler to generate and reconstruct into floating-point.

## 3.4 DESIGN CHOICE

We discuss key design choices for implementing the proposed method. Specifically, the choices outlined below enable a modular implementation, where $\widehat{W} = f(W, B_t)$ is contained within a single PyTorch module, with reasonable increase in GPU memory usage. While alternative designs could prioritize lower memory overhead over modularity, we focus here on clarity and simplicity.

**Separate kernels.** While the BF16 baseline requires only one operation for the forward pass of linear layer, the *DiffFPQ* counterpart requires three operations: (1) generating $R$, (2) unpacking $R$ and adding PQN to $W$, and (3) the matrix multiplication. Fusing consecutive operations typically helps achieve maximum throughput by reducing GPU memory communication. However, we decided not to fuse the operations, considering the following.

Firstly, $R$ generation is not fused. PRNG is an algorithm that loops based on its internal state to generate random values iteratively. The longer a PRNG's internal state is reused, the more it reduces the degree of parallelism, limiting the utilization of parallel hardware. In other words, there exists a sweet spot of parallelization that maximizes throughput. Furthermore, additional communication is required if the number of random values $R$ generated and consumed per CUDA core does not match. In practice, fusing the generation of $R$ with the subsequent operations led to significant variation of throughput depending on the shape of $W$.

Secondly, we do not fuse the PQN addition with the subsequent matrix multiplication. This decision allows us to keep implementation straightforwardly modular and to reuse the highly optimized PyTorch implementation of the linear operation.

**GPU memory.** For the gradient of input activations in matrix multiplication, $\widehat{W} = f(W, B_t, R)$ is required. While $\widehat{W}$ could be reconstructed during the backward pass without additional GPU memory overhead, we chose to reuse the value computed and stored during the forward pass. First, implementing online reconstruction in PyTorch would require fusing layers—$f(W, B_t, R)$ and the following matrix multiplication—which complicates modular implementation. Second, memory overhead of storing $\widehat{W}$ in BF16 is 2 bytes per parameter, which is manageable for small models. We focus on small models, as our experiments demand extensive training runs for $>200B$ tokens to ensure training stability.

## 3.5 IMPLEMENTATION DETAIL

**Managing seed.** A seed value is required to initialize the PRNG, and here we discuss the specific requirements for it. Note that backward computation as in Equation 2 requires $R$ which must be identical to the value of $R$ in the forward pass for proper training. Additionally, to avoid bias across the entire model, the $R$ values for each layer should be independently random.

To achieve these requirements, a multi-layer PRNG is employed to manage seeds and their corresponding random values. First, a seed generator PRNG is initialized with the user-specified seed value. Second, the seed generator is used to produce seed values to initialize the PRNG of each layer. Finally, the output of each layer's PRNG serves as the seed value for the GPU's PRNG, which then generates $R$. The state of each layer's PRNG is changed every gradient update during training.

**Bitwidth.** We implemented an internal bitwidth parameter $B_i$ for each 32 by 32 square unit of parameters in the linear layers. $B_i$ is linearly scaled to represent bitwidth $B_t$ as follows:

$$B_t = b_{\min} + B_i \cdot (b_{\text{init}} - b_{\min}) \tag{6}$$

where $b_{\text{init}}$ and $b_{\min}$ are hyperparameters representing the initial and minimum bitwidths, respectively. $B_i$ should be initialized with 1. $B_t$ is guided towards $b_{\min}$ through the weight decay applied to $B_i$. A loss term related to $B_t$ can also be incorporated into the training loss $\mathcal{L}$:

$$\mathcal{L}' = \mathcal{L} + \lambda \sum_{i=1}^{n} \frac{\sum_{j=1}^{m_i} |b_t^{i,j} - b_{\min}|}{m_i} \tag{7}$$

where $n$ is the number of layers, $m_i$ is number of square blocks in $i$-th layer and $b_t^{i,j}$ denotes bitwidth of $i$-th layer and $j$-th block of $B_t$. In this case, an additional hyperparameter $\lambda$ is required to appropriately scale the loss associated with the bitwidth parameter.

## 4 EXPERIMENTAL RESULT

Transformer (Vaswani et al., 2017)-based language models were trained from scratch: the GPT2-124M model (Radford et al., 2019) on the OpenWebText dataset (Gokaslan & Cohen, 2019), and the Llama2-{134M, 1B} models (Touvron et al., 2023) on the C4 dataset (Raffel et al., 2020). The benchmarks for inference are HellaSwag (Zellers et al., 2019) and WikiText-2 (Merity et al., 2017).

We apply the proposed method to all linear layers of all transformer blocks unless otherwise specified. 'DiffQ' represents an extension of DiffQ (Défossez et al., 2022), which is equivalent to *DiffFPQ* except for BF16 $U(-0.5, 0.5)$ in place of $\approx \lfloor \mathcal{N}(0,1)/2 \rfloor$. 'method(*I*to*M*)' denotes pre-training with the corresponding method with $b_{\text{init}} = I$ and $b_{\text{min}} = M$. The default configuration is $b_{\text{init}} = 6$, $b_{\text{min}} = 4$, AdamW optimizer (Loshchilov & Hutter, 2019) and BF16 GEMM with FP32 accumulation unless otherwise specified. Refer to Appendix E for detailed settings.

### 4.1 PRE-TRAIN RESULT

**The GPT2-124M model** is trained from scratch on the OpenWebText dataset up to 295B tokens (Karpathy, 2022). Figure 2 shows that the baseline BF16 training with a learning rate of $6 \times 10^{-4}$ proceeds smoothly whereas the counterpart with a smaller learning rate $6 \times 10^{-5}$ diverges and fails to recover. Both PQT methods mitigate such training instability while the proposed method incurs minimal increase in loss. The difference in loss between *DiffFPQ* and DiffQ is attributed to the choice of $R$. *DiffFPQ* consistently outperforms DiffQ, which aligns with the properties in Section 3.2. Specifically, non-trivial $P(R = 0)$ preserves dynamic range up to that of $fp_{e,m}\mid_{e=8,m=7}$.

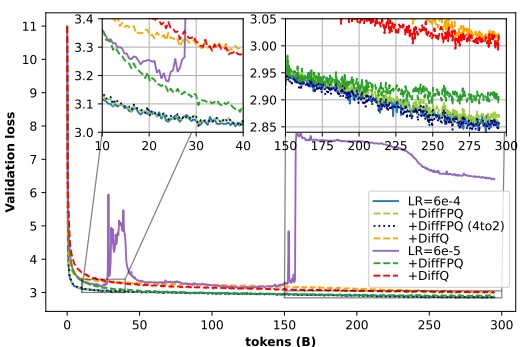

Figure 2: Loss curve of pre-training the GPT2-124M model on the OpenWebText dataset.

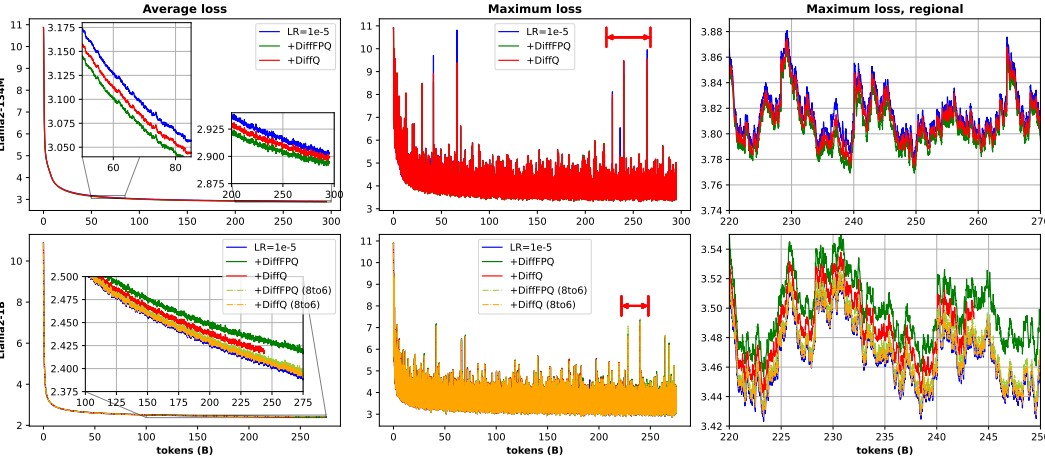

Figure 3: Training loss curve of the Llama2-{134M, 1B} models on the C4 dataset. The right column corresponds to the range annotated with the orange arrow on the middle column. The weighted moving average is used with $\alpha = 1/16$ on the left column and $\alpha = 1/128$ on the right column.

**The Llama2-134M and Llama2-1B models** are trained from scratch on the C4 dataset up to 295B and 275B tokens, respectively (Liang et al., 2025). The results are visualized in Figure 3. While PQT improves pre-training of the smaller model, it slightly degrades that of the larger model. The increase in loss with the larger model can be minimized by employing larger bitwidth hyperparameters, *e.g.,* $b_{\text{min}} = 6$. It is consistent with scaling law study in that the optimal bitwidth of larger models tends to be higher (Kumar et al., 2025). DiffQ lies in between baseline BF16 and *DiffFPQ* regardless of the model size, unlike results with the GPT2 model.

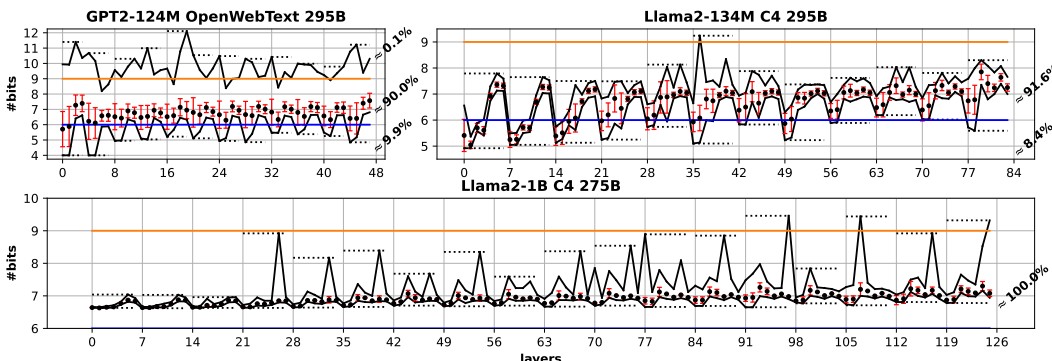

Figure 4: Resulting bitwidth $B_t$ with *DiffFPQ*(6to4) except for the Llama2-1B model which employs *DiffFPQ*(8to6). Dots and red lines indicate layerwise mean and standard deviation. Upper and lower solid lines represent layerwise maximum and minimum while dotted lines represent transformer-blockwise counterparts. Lines on 6- and 9-bit divide the parameters into 3 groups, and the percentages on the right-hand side represent the approximated ratio of parameters for each. The order of layers is (qkv, out, up, down) for GPT2 and (q, k, v, out, gate, down, up) for Llama2.

Table 2: Tokens per second per GPU and GPU memory usage during Llama2 pre-training on the A100 GPU. Subscript denotes relative overhead compared to BF16 baseline. We used local batch size $\{24, 8, 2, 2\}$ respectively for each case of $\{134M, 1B, 3B, 70B^\dagger\}$ with fixed sequence length of 2048. "$\dagger$" denotes that only 4 layers out of the total 80 layers of the model are used.

| | tps per GPU ($\times 10^3$) | | | | GPU memory (GiB) | | | |
| | **134M** | **1B** | **3B** | **70B**$^\dagger$ | **134M** | **1B** | **3B** | **70B**$^\dagger$ |
|---|---|---|---|---|---|---|---|---|
| BF16 | 143.3 | 26.0 | 7.17 | 7.22 | 34.00 | 30.69 | 19.07 | 18.83 |
| +*DiffFPQ* | $141.3_{1.40\%}$ | $25.5_{1.92\%}$ | $6.79_{5.30\%}$ | $6.94_{3.88\%}$ | 34.16 | 32.42 | 24.99 | 23.42 |
| +DiffQ | $116.6_{18.63\%}$ | $23.1_{11.15\%}$ | $5.00_{30.26\%}$ | $5.21_{27.84\%}$ | 34.18 | 32.64 | 25.76 | 25.57 |

**Resulting bitwidth** $B_t$ is visualized in Figure 4. Note that the GPT2 model results in a wider range of $B_t$ compared to the Llama2 models. It suggests that parameters of GPT2-style transformer blocks require greater dynamic range compared to Llama2-style counterparts. It is consistent with the training loss curves—*DiffFPQ* works better than DiffQ on the GPT2 model but not on the Llama2 models—with dynamic range property as in Proposition 2. On the other hand, more than 99% of the parameters are robust to PQN with $b_t \leq 9$ irrespective of the architecture and model size.

Table 2 reports the throughput and GPU memory usage during the Llama2 model training. The proposed generation method minimizes **computational overhead**. The geometric mean of the overhead on training throughput for Llama2-$\{134M, 1B, 3B, 70B^\dagger\}$ is 3.14% for *DiffFPQ* compared to 22.34% for DiffQ. On the other hand, **GPU memory overhead** is 2 bytes per parameter to store $\widehat{W}$ in BF16. Additionally, the proposed method requires less layerwise temporary memory to store $R$, using 0.5 bytes per element for $R \approx \lfloor \mathcal{N}(0,1)/2 \rceil$ compared to 2 bytes for $U(-0.5, 0.5)$.

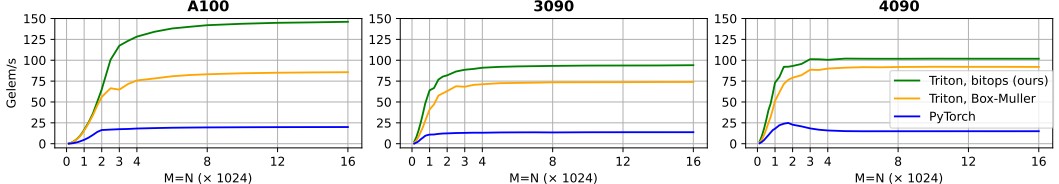

Figure 5: Forward pass benchmark results for the PyTorch layer implementing Equation 1 on a matrix $W \in \mathbb{R}^{M \times N}$ with $R \approx \lfloor \mathcal{N}(0,1)/2 \rceil$. Absolute throughput in $10^9$ elements per second.

Figure 5 reports **the results of the unit benchmark** for the forward pass of the proposed method. Both the proposed method and the Box-Muller method demonstrate at least a $3\times$ improvement compared to the PyTorch baseline, as they are implemented in Triton and reduce global memory

Table 3: Benchmark results on GPT2-124M models. For each pre-trained model, we applied FP quantization on the parameters. {FP8$^\dagger$, FP8, FP12} denote MX with internal datatypes {FP8_e4m3, FP8_e3m4, FP12_e4m7}, respectively. MP denotes mixed precision where blocks with $b_t > 6$ are quantized with FP12_e4m7, $3 < b_t \leq 6$ are quantized with FP8_e3m4, and $b_t \leq 3$ are quantized with FP4_e2m1. Specifically, the proportion of the datatypes are {FP8_e3m4 9.20%, FP12_e4m7 90.8%} (11.63-bit) for *DiffFPQ*$(6, 4)$, and {FP4_e2m1 0.628%, FP8_e3m4 16.905%, FP12_e4m7 82.466%} (11.27-bit) for *DiffFPQ*$(4, 2)$. **Bold** typefaces denote Pareto-optimal results.

| Pre-train | $(b_{\text{init}}, b_{\text{min}})$ | HellaSwag (accuracy, %) | | | | | WikiText-2 (perplexity) | | | | |
| --- | --- | --- | --- | --- | --- | --- | --- | --- | --- | --- | --- |
| | | BF16 | FP8$^\dagger$ | FP8 | FP12 | MP | BF16 | FP8$^\dagger$ | FP8 | FP12 | MP |
| BF16 | - | 31.58 | 31.58 | **31.70** | 31.57 | - | 26.21 | 26.61 | **26.21** | 26.21 | - |
| FP12 | - | 31.88 | 31.88 | 31.97 | 31.97 | - | 26.83 | 27.15 | 27.04 | 26.84 | - |
| *DiffFPQ* | $(6, 4)$ | 31.94 | **32.04** | 31.97 | **31.97** | **31.99** | 26.52 | **26.83** | 26.66 | **26.52** | 26.53 |
| | $(4, 2)$ | 31.72 | 31.74 | 31.70 | 31.74 | 31.77 | 26.51 | 26.88 | 26.55 | 26.52 | 26.53 |
| DiffQ | $(6, 4)$ | 29.42 | - | - | - | - | 33.61 | - | - | - | - |

communication. The proposed noise generation method improves throughput over the Box-Muller method across all test cases. It is particularly effective with larger matrices and the A100 GPU. Note that the weight dimension of Llama 3.2 1B ranges from $\mathbb{R}^{2048 \times 512}$ to $\mathbb{R}^{2048 \times 8192}$ while Llama 3.1 405B counterpart ranges from $\mathbb{R}^{16384 \times 1024}$ to $\mathbb{R}^{16384 \times 16384}$.

## 4.2 INFERENCE RESULT

Table 3 reports benchmark results of pre-trained models with different inference datatypes. We have GPT2-124M models pre-trained with different methods: BF16 baseline, MXFP12_e4m7, *DiffFPQ* with two different hyperparameters, and DiffQ. For each of the pre-trained models, we quantize the parameters into the corresponding datatypes without adding noise (Microsoft, 2023).

Mixed precision MXFP based on $B_t$ outperforms both FP8 variants with lower perplexity and similar accuracy. It is comparable to FP12_e4m7 with slightly higher accuracy and similar perplexity. Additionally, its data size is smaller than FP12_e4m7, achieving a 27.3% reduction for *DiffFPQ*(6,4) and 29.54% for *DiffFPQ*(4,2) relative to BF16. The results suggest that *DiffFPQ* allows PQT to target FP datatypes in Table 1 as discussed in Section 3.2.

## 4.3 ABLATION STUDY

**Efficacy of rounded distribution.** Results in Figure 6 with $R \sim \mathcal{N}(0, 1)/2$ show that the proposed rounded distribution $R \approx \lfloor \mathcal{N}(0, 1)/2 \rceil$ is indeed imperative when it comes to preserving training behavior.

**Pre-train with MXFP12_e4m7 parameters.** Following the bitwidth result that over 99% of the parameters are robust to PQN with $b_t \leq 9$, we quantize parameters to MXFP12 during training. Note that adopting MXFP12 increases the lower bound on near-zero resolution and decreases the upper bound on dynamic range compared to BF16 baseline. Figures 6 and 7 report *DiffFPQ* pre-training results with fake-quantized MXFP12 parameters, specifically MX version of $fp_{e,m}(\widehat{W}) \mid_{e=4, m=7}$. The results demonstrate that MXFP12-quantized parameters closely fol-

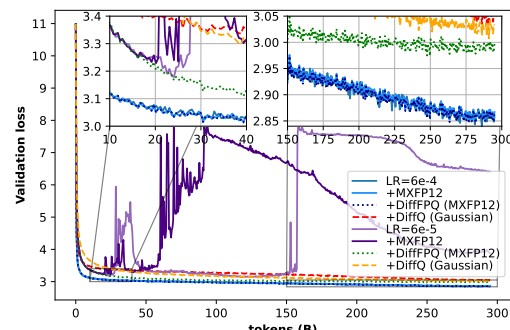

Figure 6: Ablation study with $R \sim \mathcal{N}(0, 1)/2$ (without rounding $\lfloor \cdot \rceil$, 'Gaussian') and MXFP12_e4m7 quantized parameters ('MXFP12') on the GPT2-124M model.

low their BF16 counterparts, independent of the underlying architecture and the model size, except for the GPT2-124M model with smaller learning rate. It implies that $B_t$ derived from BF16 training can suggest a baseline datatype, *i.e.,* the datatype used during training, for the model parameter $W$.

We report extended ablation studies on training stability and optimizer compatibility in Appendix D.

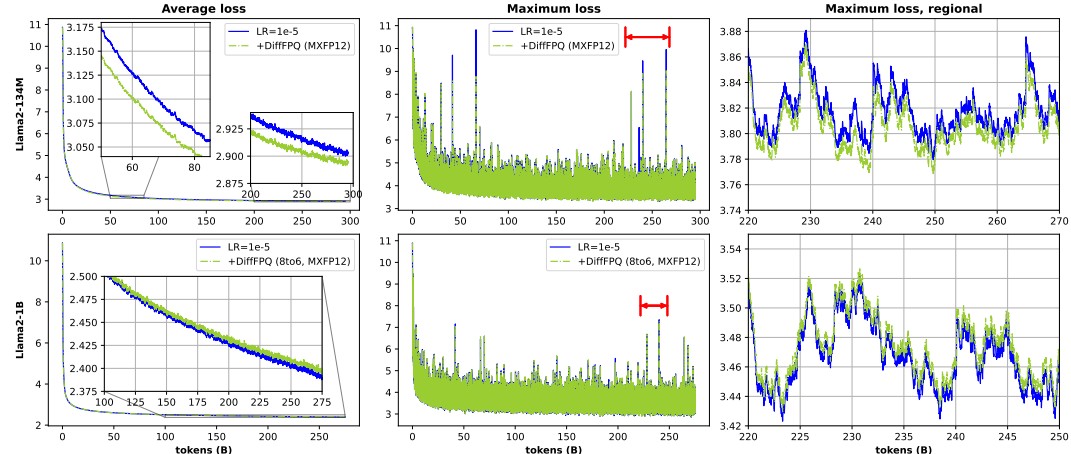

Figure 7: Ablation study on *DiffFPQ* pre-training with MXFP12_e4m7 quantized parameters on the Llama2-{134M, 1B} models.

## 5  DISCUSSION, BROADER IMPACT AND LIMITATION

The proposed method allows differentiable search over datatypes while stable and efficient training. First, *DiffFPQ* yields mixed precision datatype configuration for inference and suggests datatype for training. Second, *DiffFPQ* allows stable training irrespective of the underlying architecture and the model size. Third, *DiffFPQ* is efficient with 3.14% computational overhead over BF16, which is $7\times$ less than DiffQ.

In addition, *DiffFPQ* implicitly encourages generalization ability of the model. Note that Taylor expansion on training loss $\mathcal{L}(\widehat{W}) \approx \mathcal{L}(W) + PQN^\intercal \frac{\partial \mathcal{L}(W)}{\partial W} + PQN^\intercal \frac{\partial^2 \mathcal{L}(W)}{\partial W^2} PQN$ includes Hessian trace so that optimization on $\mathcal{L}(\widehat{W})$ converges to flatter minima (Shin et al., 2023). Our results—that *DiffFPQ*-trained model yields higher accuracy with slight increase in perplexity—imply that PQT with the proposed $R \approx \lfloor \mathcal{N}(0,1)/2 \rceil$ encourages generalized learning via flatter minima.

We further discuss implications of the proposed method. First, *DiffFPQ* can replace mixed-precision QAT methods or even vanilla BF16 training. Compared to the mixed-precision QAT method employed in Gemma 3n, *DiffFPQ* allows a wider range of datatypes with time-resource overhead disentangled from the number of datatype options (Sanseviero & Ballantyne, 2025; Nair et al., 2025). Compared to BF16 training, *DiffFPQ* offers stable and mixed-precision training that encourages generalization ability of the model, while incurring minimal computational overhead. Therefore, we believe that *DiffFPQ* to be an appealing option for mixed-precision foundational models.

Second, *DiffFPQ* provides theoretical ground on specific configuration of FP datatypes, at least for model parameters. In terms of inference, *DiffFPQ*-trained models achieve Pareto-optimal benchmark results with optimal MXFP-quantized parameters. On the other hand for training, MXFP12_e4m7—following that more than 99% of parameters are robust to PQN with $b_t \leq 9$—demonstrates to be capable of representing the parameters similar to BF16 counterpart irrespective of the underlying architecture and model size. Furthermore, Propositions 1 and 2 are general enough to be applied to other MX-like formats such as NVFP4.[4] Therefore, we hope the proposed method will serve as a theoretical foundation for FP standards. Note that, to the best of authors' knowledge, there has been no well-defined theoretical ground on the specific configuration of FP, *e.g.,* 5-bit exponent for half-precision over others (IEEE, 2019; Project, 2023).

The proposed method is applied only on weight, leaving activation and gradient the same as baseline BF16. In particular, it is impossible to conduct a differentiable search on gradients. Extending the proposed method to activation is left as future work. The use of LLMs during this work is limited to (1) suggesting academic rewrite for the contents of the paper and (2) writing code for visualization.

---

[4] NVFP4 utilizes shared scale in FP8_e4m3 format, making the quantization range not exactly power-of-two. Therefore, PQT targeting NVFP4 requires fake-quantization with appropriate scale, unlike its MX counterpart.

## REPRODUCIBILITY STATEMENT

We believe that the paper contains enough information to reproduce the results. First, Section 3 explains how to implement the proposed method. Specifically,

- Section 3.1 explains square-blockwise quantization and forward-backward formulation.
- Section 3.3 describes the distribution $R$ as in Equation 5 and the base cases to generate such distribution as in Equation 4.
- Section 3.4 explains the design choices for the implementation.
- Section 3.5 explains implementation details for internal bitwidth parameter $B_i$ as in Equation 6 and optional bitwidth loss as in Equation 7.

Second, Sections 4 and E explain the specific experimental setup down to the level of container image and hyperparameter settings. Lastly, we provide the source code, which is version-controlled using git, for a reviewing purpose.

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

## A  PROOF

Proposition 1

*Proof.* Consider adding $PQN_i$ on a near-zero element $\epsilon_i \in W$. Without loss of generality, consider a single block and the smallest positive perturbation $\epsilon_{PQN}$ with $\min_{>0}(|R|) = 2^\rho$ that limits the precision of near-zero floating-point $\epsilon_i$ in the least. The value of interest is:

$$\epsilon_i + \epsilon_{PQN} = \epsilon_i + 2^{\rho+1-b_t} \max(|W|) \tag{8}$$

Note that the resolution of FP depends on power-of-two magnitude of the data, *e.g.*, $\Delta_{\min}(\epsilon_{PQN}) = 2^{\lfloor s \rfloor - m}$ where $s \triangleq \rho + 1 - b_t + \log_2 \max(|W|)$. Also note that FP addition $\epsilon_i + \epsilon_{PQN}$ implies FP casting, *e.g.*, $fp_{e,m} \mid_{e=8,m=23} (\epsilon_i + \epsilon_{PQN})$ for single precision. Therefore, the minimum effective delta of $\epsilon_i$ with respect to $\epsilon_i + \epsilon_{PQN}$ is limited:

$$\min_{>0}(|fp_{e,m}(\epsilon_i + \epsilon_{PQN}) - fp_{e,m}(\epsilon_{PQN})|) = \Delta_{\min}(\epsilon_{PQN}) \tag{9}$$

unless $\epsilon_{PQN}$ is exact power-of-two such that $\exists n \in \mathbb{Z} \; \epsilon_{PQN} = 2^n$, and $\epsilon_i < 0$. That is, the addition $\epsilon_i + \epsilon_{PQN}$ limits effective resolution of $\epsilon_i$:

$$\Delta_{\text{eff}}(\epsilon_i) \geq \Delta_{\min}(\epsilon_{PQN}) \tag{10}$$

Given limited effective resolution of $\epsilon_i$, FP with limited number of exponent ranges where $\Delta_{\min}(x) \geq 2^{\lfloor s \rfloor - m} \; \forall x$ can represent $W$. It corresponds to an FP where $[2^{\lfloor s \rfloor}, 2^{\lfloor s \rfloor + 1})$ is the smallest normal range. We can count the number of effective exponent ranges from the largest exponent range to the smallest exponent range. There are $(-\rho + b_t)$ exponent ranges between the largest exponent range

$$[2^{\lfloor \log_2 \max(|W|) \rfloor}, 2^{\lfloor \log_2 \max(|W|) \rfloor + 1}) \tag{11}$$

and the smallest exponent range

$$[2^{\lfloor \rho + 1 - b_t + \log_2 \max(|W|) \rfloor}, 2^{\lfloor \rho + 1 - b_t + \log_2 \max(|W|) \rfloor + 1}) \tag{12}$$

including both endpoints. Considering a single exponent range for subnormal and no dedicated exponent range for inf/NaN, FP with a $\lceil \log_2(-\rho + b_t + 1) \rceil$-bit exponent can represent $W$ effectively. $\square$

Proposition 2

*Proof.* Assume $\exists n \in \mathbb{N}^+$ such that $\tau_k = 2^{-n} \cdot \epsilon_{PQN}$. For any $i \in \{1, 2, \ldots, k\}$, if $R_i \neq 0$, $|\epsilon_i + PQN_i| > (1 - 2^n) \cdot \epsilon_{PQN}$ holds because:

- $\epsilon_i + \epsilon_{PQN}$ is in the range $((1 - 2^{-n}) \cdot \epsilon_{PQN}, (1 + 2^{-n}) \cdot \epsilon_{PQN})$.
- $\epsilon_i - \epsilon_{PQN}$ is in the range $((-1 - 2^{-n}) \cdot \epsilon_{PQN}, (-1 + 2^{-n}) \cdot \epsilon_{PQN})$.

On the other hand, $R_i = 0$ yields $PQN_i = 0$ and $\widehat{W}_i = W_i$. If $\exists i \in \{1, 2, \ldots, k\}$ such that $R_i = 0$, $|\epsilon_i + PQN_i| = |\epsilon_i| < \tau_k$. Such an element becomes an upper bound on $\min_{>0}(|\widehat{W}|)$.

The condition $\min_{>0}(|\widehat{W}|) < 2^{-n} \cdot \epsilon_{PQN}$ holds if $\exists i \in \{1, 2, \ldots, k\}$ such that $R_i = 0$, whose probability is $1 - (1-p)^k$. Note that the probability is not complete for the condition. The condition also holds if $\exists i$ such that $R_i \neq 0$, $|W_i| > \tau_k$ and $|W_i + PQN_i| < 2^{-n} \cdot \epsilon_{PQN}$. Therefore, the probability is lower bounded by $1 - (1-p)^k$:

$$P\left(\min_{>0}(|\widehat{W}|) < \tau_k\right) \geq 1 - (1-p)^k \tag{13}$$

$\square$

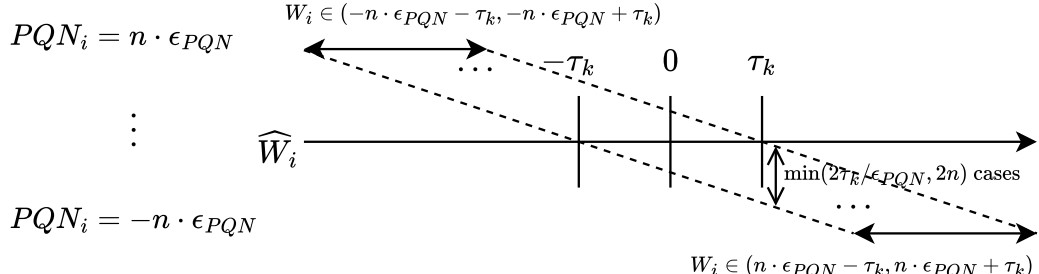

$$PQN_i = n \cdot \epsilon_{PQN}$$

$$\vdots$$

$$PQN_i = -n \cdot \epsilon_{PQN}$$

Figure B.1: Enumerative cases of $\widehat{W}_i = W_i + PQN_i$. Trapezoidal range represents cases that the addition results in $\widehat{W}_i \in (-\tau_k, \tau_k)$ with respect to different $PQN_i$ on the left. For arbitrary $|W_i| < n \cdot \epsilon_{PQN} + \tau_k$, there are at most $\min(2\tau_k/\epsilon_{PQN}, 2n)$ cases of $PQN_i$ that results in $\widehat{W}_i \in (-\tau_k, \tau_k)$.

## B  DYNAMIC RANGE WITH NONZERO $R$

Consider $\min_{>0}(|\widehat{W}|) = \widehat{W}_i \mid_{PQN_i \neq 0}$.

**Proposition 3.** *Assume there are $k$ elements of $W$ such that $|W_i| < \tau_k$. Further assume that there are $k_2$ elements of $W$ such that $|W_i| < \tau_k + \upsilon_{PQN}$ where $\upsilon_{PQN} = \max(|PQN|) > \tau_k$ with $R \sim U(-0.5, 0.5)$, $P(R = 0) = 0$ and $\tau_k < \upsilon_{PQN}$. Then,*

$$P\left(\min_{>0}(|\widehat{W}|) < \tau_k\right) \leq 1 - \left(1 - \frac{\tau_k}{\upsilon_{PQN}}\right)^{k_2} = 1 - \left(1 - \frac{\tau_k}{\max(|W|)}2^{b_t}\right)^{k_2} \tag{14}$$

*Proof.* Assume that any element of PQN is one of $\{\pm\epsilon_{PQN}, \pm2\epsilon_{PQN}, \ldots, \pm n \cdot \epsilon_{PQN}\}$ where $n \cdot \epsilon_{PQN} = \upsilon_{PQN}$. For $\widehat{W}_i \in (-\tau_k, \tau_k)$ to hold, $W_i \in (-\tau_k - c \cdot \epsilon_{PQN}, \tau_k - c \cdot \epsilon_{PQN})$ should hold with $PQN_i = c \cdot \epsilon_{PQN}$ where $c \in \{\pm1, \pm2, \ldots, \pm n\}$. Trapezoidal range as in Figure B.1 represents such cases.

Consider the condition that arbitrary $-\tau_k - \upsilon_{PQN} < W_i < \tau_k + \upsilon_{PQN}$ results in $\widehat{W}_i \in (-\tau_k, \tau_k)$. There are at most $\min(2\tau_k/\epsilon_{PQN}, 2n)$ cases of PQN for the condition to hold. As we assume $R \sim U(-0.5, 0.5)$ and $\tau_k < \upsilon_{PQN}$, the probability is:

$$\frac{\tau_k}{n \cdot \epsilon_{PQN}} = \frac{\tau_k}{\upsilon_{PQN}} = \frac{\tau_k}{\max(|W|)}2^{b_t} \tag{15}$$

The condition $\min_{>0}(|\widehat{W}|) < \tau_k$ holds if any one of the $k_2$ elements results in $\widehat{W}_i \in (-\tau_k, \tau_k)$. Therefore,

$$P\left(\min_{>0}(|\widehat{W}|) < \tau_k\right) \leq 1 - \left(1 - \frac{\tau_k}{\upsilon_{PQN}}\right)^{k_2} = 1 - \left(1 - \frac{\tau_k}{\max(|W|)}2^{b_t}\right)^{k_2} \tag{16}$$

$\square$

As per Propositions 2 and 3, we can compare $R \approx \lfloor \mathcal{N}(0, 1)/2 \rceil$ to $R \sim U(-0.5, 0.5)$ with respect to $\min_{>0}(|\widehat{W}|)$. For example, for the 32 by 32 square block units of the pre-trained GPT2-124M model, $\frac{\tau_k}{\max(|W|)}$ on average is approximately $2^{-10}$ for $k = 3$, and $2^{-9}$ for $k = 5$. Assuming $k = 3$ and $k_2 = 5$, the probability with $R \approx \lfloor \mathcal{N}(0, 1)/2 \rceil$ is approximately $0.977$ independent of $b_t$. On the other hand, the probability with $R \sim U(-0.5, 0.5)$ is approximately $0.487$ with $b_t = 6$ and $0.276$ with $b_t = 5$. Therefore, $R \approx \lfloor \mathcal{N}(0, 1)/2 \rceil$ is much more likely to preserve the minimum magnitude, and thereby dynamic range, of $W$ compared to $R \sim U(-0.5, 0.5)$. Also note that $\min_{>0}(|\widehat{W}|)$ with $R \approx \lfloor \mathcal{N}(0, 1)/2 \rceil$ is pass-through of high-precision $W_i$ with $R_i = 0$ while the counterpart with $R \sim U(-0.5, 0.5)$ has limited resolution of $\max(\Delta_{\max}(|W_i|), \Delta_{\max}(|PQN_i|))$.

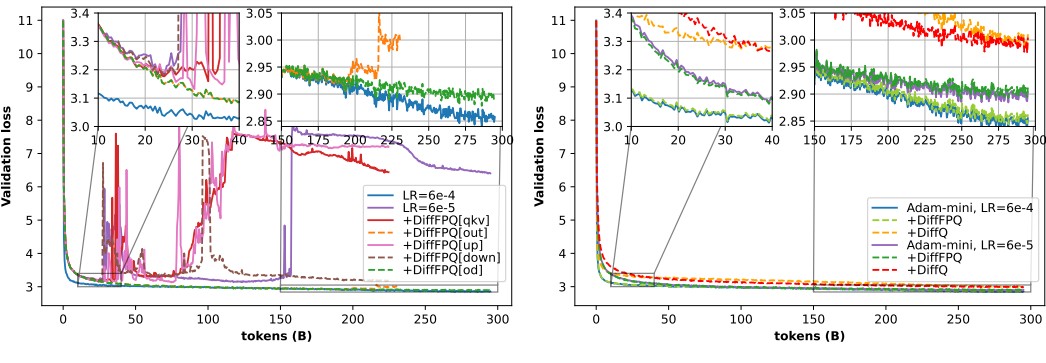

Figure C.1: An example of vector-wise quantization on $W_{(K,N)} \sim \mathcal{N}(0,1)$ and its forward-backward discrepancy, where $K = N = 4$. Boxes wrapped in bold solid lines represent quantization groups with an internal datatype of INT4 and a block size of 2. Visualized values are fake-quantized.

## C TRANSPOSE-COMMUTATIVITY

A naïve application of low-precision training can compromise the consistency of values between the forward and backward passes, which hurts training stability. Consider forward and backward passes of an MX-compliant matrix multiplication where the quantization axis lies along the inner dimension:

$$T_{(M,N)} = A_{(M,K_Q)} W_{(K_Q,N)} \tag{17}$$

$$\frac{\partial L}{\partial W}_{(K,N)} = A^{\mathsf{T}}_{(K,M_Q)} \frac{\partial L}{\partial T}_{(M_Q,N)} \quad \text{and} \quad \frac{\partial L}{\partial A}_{(M,K)} = \frac{\partial L}{\partial T}_{(M,N_Q)} W^{\mathsf{T}}_{(N_Q,K)} \tag{18}$$

where $A$ denotes the input activation, $W$ the parameter, $T$ the output activation and $L$ the target loss. The subscript corresponds to the shape of the matrix and $_Q$ denotes quantization axis. Note the difference of $W$ between the forward and backward passes, *i.e.*, $W_{(K_Q,N)}$ compared to $W^{\mathsf{T}}_{(N_Q,K)}$, which is demonstrated in Figure C.1. This inconsistency can lead to suboptimal training (Chen et al., 2025). The issue arises because the absolute maximum value of the block, *e.g.*, size-2 blocks in Figure C.1, changes when transposed. Square-blockwise quantization resolves this problem by ensuring transpose-commutativity.

## D EXTENDED ABLATION STUDY

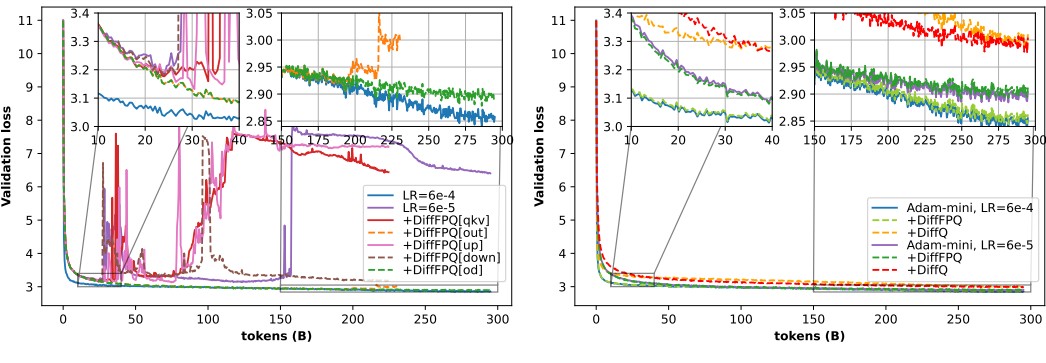

(a) Stability case study with $b_{\min} = 0$.      (b) Orthogonal to the choice of optimizer Adam-mini.

Figure D.1: Training loss curve of the GPT2-124M model on the OpenWebText dataset.

**Stability case study.** To identify the source of baseline BF16 training instability, we restrict the application of the proposed method to each of the linear layers within all transformer blocks. '*DiffFPQ*[part]' denotes which layer(s) of all transformer blocks adopt the proposed method, where [od] is used as shorthand for [out,down]. Note that the GPT2 transformer block comprises four linear layers: qkv, out, up, and down. The qkv and out layers, along with the self-attention operation, constitute the attention module, while the up and down layers form the feed-forward module.

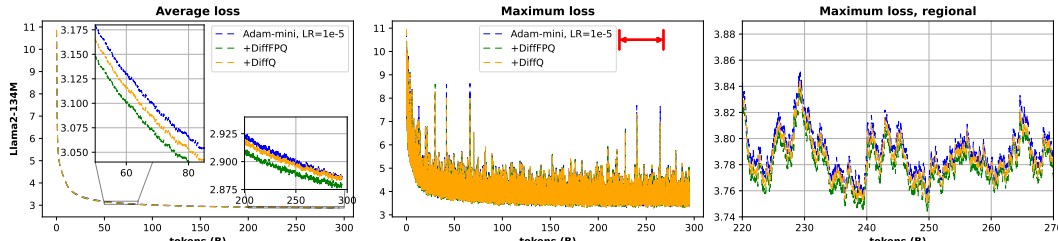

Figure D.2: Training loss curve of the Llama2-134M model on the C4 dataset with Adam-mini optimizer. The rightmost column corresponds to the range annotated with the orange arrow on the second column. For better visualization, weighted moving average is used with $\alpha = 1/16$ on the left column and $\alpha = 1/128$ on the right column.

Table E.1: Hyperparameters used for pre-training.

|  | GPT2-124M | Llama2-134M | Llama2-1B |
|---|---|---|---|
| context window | 1024 | 2048 | 2048 |
| local batch | 12 | 12 | 8 |
| local grad. accum. per step | 5 | 1 | 1 |
| # GPUs | 8 | 8 | 8 |
| total steps | 600k | 1.5M | 2.1M |
| warmup steps | 2k | 5k | 7k |
| total number of training tokens ($\times 10^6$) | 294,912 | 294,912 | 275,251.2 |
| min learning rate | {6e-5, 6e-6} | 1e-6 | 1e-6 |
| max learning rate | {6e-4, 6e-5} | 1e-5 | 1e-5 |
| weight decay | 0.1 | 0.1 | 0.1 |
| $\lambda$ (as in Equation 7) | 1e-4 | 0 | 0 |

As shown in Figure D.1a, *DiffFPQ*[qkv], *DiffFPQ*[up], and *DiffFPQ*[down] begin to diverge at $\approx 30$B tokens of training and fail to recover. In contrast, *DiffFPQ*[out] does not diverge and closely approximates the best-case loss curve of baseline BF16 up to $\approx 200$B tokens. *DiffFPQ*[od], which applies the proposed method to the last layers of the residual addition branches in the transformer blocks, reduces divergence and yields the best result with the smaller learning rate. These results show that the attention module is the source of instability at $\approx 30$B tokens of training, while the feed-forward module is the source of instability at $\approx 200$B tokens of training. The latter is consistent with Fishman et al. (2025).

**Adam-mini.** Pre-training results with the Adam-mini optimizer (Zhang et al., 2025) are presented in Figures D.1b and D.2. Results with *DiffFPQ* require fewer tokens for Adam-mini to surpass AdamW. *DiffFPQ* is orthogonal to the choice of optimizer, at least for Adam-mini.

# E   PRE-TRAIN HYPERPARAMETER, SETUP AND RESOURCE

Table E.1 reports hyperparameters used for pre-training. Learning rate was linearly scheduled with warmup. The Llama2-1B model training requires more than 24GiB of GPU memory.

For GPT2, we used Karpathy (2022) with commit `9755682b` as a starting point and `nvcr.io/nvidia/pytorch:24.10-py3` with Triton v3.2.0 as a training environment. For Llama2, we used Liang et al. (2025) with commit `90567fc9` as a starting point and `ghcr.io/pytorch/pytorch-nightly` with a tag `2.7.0.dev20250107-cuda12.4-cudnn9-devel` as a training environment. We used A100-SXM4-40G, RTX 3090 and RTX 4090 GPUs with NVIDIA driver R565.

