# OpenReview forum: "Differentiable, Stable and Efficient Floating-Point Quantization"
_ICLR.cc/2026/Conference — ICLR 2026 Conference Withdrawn Submission_

### Official Review · Reviewer_ezBK · 2025-10-25

**Soundness:** 1
**Presentation:** 1
**Contribution:** 2
**Rating:** 2
**Confidence:** 2

**Summary:**

This paper proposes a pseudo-quantization training (PQT) method called DiffFPQ that can differentiably optimize on the microscaling (MX) datatypes for LLM quantization.

**Strengths:**

In terms of originality, this paper introduces a novel pseudo-quantization noise (PQN) that can help to optimize the floating-point (FP) datatypes instead of the integer datatypes. It also provides Triton-based GPU kernels to efficiently generate PQN during model pre-training.

In terms of quality, both theoretical and experimental results are presented in the paper. The source code is provided in the supplementary material.

In terms of clarity, the motivation for doing PQT on MX datatypes is well-described.

In terms of significance, the proposed method (DiffFPQ) showed similar or better training loss convergence and evaluation accuracies compared to the baseline and the other method, DiffQ.

**Weaknesses:**

Disclaimer: I apologize in advance if I have any misunderstandings. However, I do not think this paper is well-written, and many details are not explained in the paper. I have tried to read this paper several times, but it is still hard to understand.

1. The formula in Equation 1 is not well-motivated for MX datatypes. I can see this is an analog to the DiffQ: $\hat{W} = W + R \cdot \Delta$. While it makes a lot of sense in integer quantization in DiffQ, it does not look intuitive for MX datatypes.
For integers, the step size $\Delta$ is fixed and is proportional to the maximum element within a group, so it is independent of the elements that are not the maximum. By setting $R \sim U(-0.5, 0.5)$, this simulates well the effect of round-to-nearest.
For FP datatypes, however, the step size is not fixed within a group and is more or less proportional to each element, regardless of whether the element is the maximum or not. So a formula with the form like $\hat{W} = W + R \cdot \Delta \cdot W$ should be more appropriate.

2. In Section 3.3, the connection between Equations 4 and 5 is not clear. The algorithm to generate $R$ using bitwise operations is not described.

3. In Figure 4, most of the blocks have the bitwidth larger the $b_\mathrm{min}=4$ and even larger than $b_\mathrm{init}=6$. but by Equation 7, the bitwidth should be guided to decay from $b_\mathrm{init}$ to $b_\mathrm{min}$. This is inconsistent.

4. Despite mentioning the term Pareto-optimal several times, the paper does not explain what Pareto-optimal means in the context of PQT. Several numbers as marked in bold in Table 3 to denote Pareto-optimal, but it is not clear why they are Pareto-optimal.

5. This paper lacks the practical deployment information for inference. In particular, Figure 4 shows that the blocks within the same layer have different bitwidths, which is not ideal for hardware computation. There is no information on the computational speed or memory usage for inference.

**Questions:**

1. In Equation 2, the symbol $\sum_{bl}$ seems undefined. Does it mean to sum the square blocks?

2. In Figure 1(b), what do the left and right arrows represent? And what do the small segments below the arrows represent?

3. In Line 147, what is non-trivial $P (R = 0)$, and what is trivial $P (R = 0)$?

4. What does the "relatively trivial" mean in Line 154?

5. In Equation 7, why do you divide $m_i$ (the number of blocks in the $i$-th layer) for the bitwidth decay loss? I think each block should contribute equally to the total size of the model, regardless of which layer it is located in.

6. In Line 298, how does it preserve $fp_{e,m}|{e=8,m=7}$, aka BF16, if the bitwidth $b_\mathrm{init}=6$?

7. In Figure 3, why do DiffFPQ and DiffQ converge faster than the LR=1e-5 baseline when adding the PQN noise?

8. In Line 361, why do you use the geometric mean instead of the more commonly used arithmetic mean when calculating the computational overhead in pre-training?

9. In Table 3, under what condition of $b_t$ is the datatype FP8$^\dagger$ (FP8_e4m3) being used? Where are the FP4_e2m1 results?

---

### Official Review · Reviewer_q7zi · 2025-10-29

**Soundness:** 3
**Presentation:** 2
**Contribution:** 2
**Rating:** 4
**Confidence:** 3

**Summary:**

This paper introduces DiffFPQ, a differentiable, stable, and efficient floating-point quantization (FPQ) method for neural network training. The core idea is to employ Pseudo-Quantization Noise (PQN), sampled from a rounded normal distribution $R \approx \lfloor \mathcal{N}(0,1)/2 \rceil$, to construct differentiable objectives across mixed-precision floating-point configurations. This method enables the optimization of FP bit configurations as continuous parameters while preserving the dynamic range of weights. Experimental validation is provided on GPT2-124M and Llama2-{134M, 1B} models trained up to 295B tokens, demonstrating stable and efficient training, benchmarking against existing methods (DiffQ, BF16), and offering insights into bit-width allocation and the corresponding optimal MXFP data types.

**Strengths:**

1.  **Methodology is Relatively Novel and Logically Efficient from a Systems Perspective:** The paper utilizes a Pseudo-Quantization Noise (PQN) formulation based on rounded normal noise ($R \approx \lfloor\mathcal{N}(0,1)/2\rceil$) to generalize pseudo-quantization training from integer data types to floating-point data types, enabling differentiable optimization over mixed-precision FP configurations. This effectively reduces the exponential search space inherent in quantization to a manageable form.
2.  **Algorithm is Efficient, with Reproducibility Claims:** The proposal to replace floating-point random number generation with bit-operations for generating $R$ significantly reduces training overhead, as demonstrated in Table 2 (Section 4.1) and Figure 5 (unit throughput benchmarks). The overhead is quantified and shown to be negligible (3.14%) compared to previous methods (e.g., DiffQ).

**Weaknesses:**

1.  **Presentation is Insufficient and Difficult to Read:** Certain sections (particularly Section 3.2) are overly concise and technical, lacking explanations that are easy to understand. More intuition, motivation, and broader background would enhance readability for non-experts. Furthermore, there's a lack of intuitive connection between the theoretical proofs in Section 3 and the experimental conclusions in Section 4, making it unclear for readers how the theoretical improvements translate into practical benefits.
2.  **The Main Claim of the Paper is Unclear:** It's difficult to determine if the paper aims to solve the problem of insufficient accuracy in current pseudo-quantization training (PQT) or to more broadly define an optimal quantized data format directly for quantized training. Regardless of the title, introduction, or conclusion, the impression given is not solely about improving PQT. However, if the primary motivation is the latter, the paper lacks solid arguments to derive the optimal quantization format and lacks comparisons with current mainstream quantization training frameworks (both in terms of system speed and training accuracy).
3.  **The claim made in the Discussion section that "DiffFPQ can replace mixed-precision QAT methods or even vanilla BF16 training" is unconvincing.** While the DiffFPQ method does offer good system-level throughput optimization, it (1) focuses only on weight quantization and (2) does not address the training effectiveness in extremely low-bit quantization scenarios. This differs significantly from the research content and application prospects of current mainstream QAT papers. At best, it can provide system-level inspiration for QAT, but cannot replace it.

**Questions:**

1.  Can the authors provide further details on the benefits that the search logic proposed by the DiffFPQ method brings at the end-to-end level?
2.  Will DiffFPQ generalize to activation and/or gradient quantization, and what challenges are anticipated in extending the method? What would be the expected effects (empirically or theoretically)?

---

### Official Review · Reviewer_5BoY · 2025-10-29

**Soundness:** 3
**Presentation:** 3
**Contribution:** 2
**Rating:** 6
**Confidence:** 2

**Summary:**

This paper proposes DiffFPQ, a differentiable pseudo-quantization training framework targeting MXFP (power-of-two, blockwise) floating formats. It adds per-block pseudo-quantization noise scaled by the block max and sampled from a rounded Gaussian, enabling stochastic precision annealing that preserves dynamic range while training robustness to low bit-widths. From-scratch pretraining on GPT-2-124M and LLaMA2-{134M, 1B}  shows stable training with ≈3% throughput overhead(vs BF16) and improvements over a uniform-noise PQT baseline.

**Strengths:**

1. Introduces a differentiable pseudo-quantization scheme tailored to MXFP with a principled “rounded-Gaussian” noise that preserves dynamic range via stochastic precision annealing.
2. Bit-ops–based noise sampling and modular kernels keep training overhead very low (3%) while avoiding fragile kernel fusion.
3. Demonstrates stable pretraining from scratch across multiple model sizes and architectures, with sensible behavior when widening the target bit-width.

**Weaknesses:**

1. The method leaves activations and gradients in BF16 and caches W_c ​ in BF16 during training. For large models, this adds extra memory footprint, which may blunt end-to-end efficiency gains.
2. Mixed-precision MXFP quality is only reported on GPT-2-124M and two small benchmarks; there’s no Llama2 inference table or task diversity to establish external validity under the same quantization pipeline.

**Questions:**

1. Beyond HellaSwag and WikiText-2, can you report zero-shot (and, if feasible, 5-shot) results for both GPT-2-124M and Llama2-{134M,1B} on a wider NLP suite—e.g., ARC-E, ARC-C, PIQA, BoolQ, Winogrande, OpenBookQA, and selected MMLU subsets—using the same MP-MXFP quantization pipeline, and decoding setting as the baselines, with confidence intervals to assess external validity?

2. Can you report results for unmodified DiffQ—original​, the published grouping, and a sweep over the size/bit-penalty—under the same token budget and data? This would clarify whether Table 3’s DiffQ degradation is due to the method or the configuration.

---

### Official Review · Reviewer_F86H · 2025-10-31

**Soundness:** 3
**Presentation:** 3
**Contribution:** 1
**Rating:** 2
**Confidence:** 3

**Summary:**

Authors propose pseudo-quantization noise (PQN) based on R ≈ ⌊N (0, 1)/2⌉.
It allows PQT to (1) optimize on the floating-point (FP) bit configuration, (2) help
preserve dynamic range of original data, and (3) generate noise R efficiently.
They demonstrate that the proposed method allows stable and efficient pre-training
of the GPT2 and Llama2 language models up to 1 billion (B) parameters.

**Strengths:**

Authors proposed to use pseudo-quantization noise (PQN) based on R ≈ ⌊N (0, 1)/2⌉, (instead of uniform noise used for int quantization in the past) which is better fit for float quantization.
They showed that pseudo-quantization noise (PQN) based on R ≈ ⌊N (0, 1)/2⌉, is better than uniform noise (previously used for uniform int quantization).

**Weaknesses:**

Experiment is produced on small LLM with 124M parameter. It is better to use larger model size, so that downstream task accuracy can be evaluated with good confidence.
HellaSwag accuracy is in range of 30%, it is not that far from random guess 25%, it is not clear what is the confidence interval for the reported accuracy.

Comparing DiffFPQ vs DiffQ is not fair: DiffQ is designed for uniform int quantization.
So fair comparison could be apply DiffQ but use int quantization with the same budget of FP quantization. E.g. if FP4 is used for DiffFPQ then DiffQ can use int4.

Proposed method is applied for weights only quantization. There are several methods are published for FP weights only quantization, and it would be great to do a fair comparison with SOTA FP4 quantization methods e.g.:
Optimizing Large Language Model Training Using FP4 Quantization.

FP4 type is used only on 0.628% and mostly (82.466%) FP12 is used in mixed precision results. So in this case we can call the model 11 bits quantized.
Authors also present FP8 quantization results. But it is not SOTA in terms of model compression.
For example post training quantization (paper LLM-FP4: 4-Bit Floating-Point Quantized Transformers) is already showing compelling results on FP4 quantization. I would expect stronger results from quantization aware training.

**Questions:**

It is not clear what is the value of searching for mixed precision weights only quantization, given that the best it could find is 11 bits (Table 3).
Why there is no results for FP4 quantization (e.g. 90% of the model quantized with FP4)?

---

### Note · Authors · 2025-11-14

**Comment:**

Thank you to the reviewers for taking your time and effort into our paper.

Especially, comments from reviewers `5BoY` and `q7zi` turned out to be helpful for us. We note the following points:
- Unclear main claims
- Lack of extensive benchmarks

We will go on with our next steps to improve the proposed method.

On the other hand, we decided to withdraw our submission because:
- We understand that the submission requires major revision to:
  - clarify main claims
  - improve readability
- We cannot prepare additional experiments in time.

For those who have any questions on the paper, feel free to send an email.
We will be more than happy to discuss the related topics.

**Withdrawal Confirmation:**

I have read and agree with the venue's withdrawal policy on behalf of myself and my co-authors.